# Review of Medicinal Plants and Active Pharmaceutical Ingredients against Aquatic Pathogenic Viruses

**DOI:** 10.3390/v14061281

**Published:** 2022-06-13

**Authors:** Wenyu Liao, Lin Huang, Shuyu Han, Dasheng Hu, Youhou Xu, Mingzhu Liu, Qing Yu, Shuaishuai Huang, Dongdong Wei, Pengfei Li

**Affiliations:** 1Guangxi Key Laboratory of Aquatic Biotechnology and Modern Ecological Aquaculture, Guangxi Engineering Research Center for Fishery Major Diseases Control and Efficient Healthy Breeding Industrial Technology (GERCFT), Guangxi Academy of Sciences, Nanning 530007, China; lwy15805191370@163.com (W.L.); hlin556@163.com (L.H.); henry0779@163.com (S.H.); hudasheng@tom.com (D.H.); liumz1988@126.com (M.L.); yu_qing1990@163.com (Q.Y.); 13797226993@163.com (S.H.); 2Guangxi Key Laboratory of Beibu Gulf Marine Biodiversity Conservation, College of Marine Sciences, Beibu Gulf University, Qinzhou 535000, China; xyh_2016@bbgu.edu.cn; 3Guangxi Fisheries Technology Extension Station, Nanning 530007, China

**Keywords:** aquaculture, medicinal plants, pharmaceutical ingredients, chemical drugs, antiviral activities, mechanism

## Abstract

Aquaculture offers a promising source of economic and healthy protein for human consumption, which can improve wellbeing. Viral diseases are the most serious type of diseases affecting aquatic animals and a major obstacle to the development of the aquaculture industry. In the background of antibiotic-free farming, the development and application of antibiotic alternatives has become one of the most important issues in aquaculture. In recent years, many medicinal plants and their active pharmaceutical ingredients have been found to be effective in the treatment and prevention of viral diseases in aquatic animals. Compared with chemical drugs and antibiotics, medicinal plants have fewer side-effects, produce little drug resistance, and exhibit low toxicity to the water environment. Most medicinal plants can effectively improve the growth performance of aquatic animals; thus, they are becoming increasingly valued and widely used in aquaculture. The present review summarizes the promising antiviral activities of medicinal plants and their active pharmaceutical ingredients against aquatic viruses. Furthermore, it also explains their possible mechanisms of action and possible implications in the prevention or treatment of viral diseases in aquaculture. This article could lay the foundation for the future development of harmless drugs for the prevention and control of viral disease outbreaks in aquaculture.

## 1. Introduction

With the increasing demand for high-quality food in modern society, aquatic products account for an increasing proportion of meat consumption [1], because modern nutrition believes that excessive intake of red meat increases the risk of rectal cancer, advocating its replacement with other foods [2], while fish and shrimp foods are also able to provide the body with essential amino acids, lipids, and minerals, representing an excellent substitute [3]. With the increasing tension in global aquatic wild fishing resources and the increasing demand for aquaculture production, the aquaculture industry has been developing rapidly while facing increased challenges, especially the steep increase in the outbreak of aquatic animal diseases due to intensive farming, resulting in the death of a large number of farmed species and serious economic losses [4,5]. More importantly, antibiotics cannot control viral diseases; when aquatic animals are infected with viral diseases, the use of antibiotics by unprofessional farmers not only fails to alleviate the condition, but also exacerbates water pollution, resulting in an increase in mortality instead of a decrease [6].

Studies have shown that only 20–30% of antibiotics can be absorbed by farmed fish, with most of them entering the water environment [7]. The addition of antibiotics to fish and shellfish production sites via feed is a direct route to aquatic environmental pollution [8]. Intensive aquaculture leads to the overfeeding of aquatic animals and the flow of large amounts of antibiotics from ponds to lakes and then to the oceans, contributing to the deterioration of global biodiversity [9]. Abuse of antibiotics can easily cause irreversible consequences in the ecosystem. More seriously, the harmful components of antibiotic and chemical agents will remain in aquatic products, and then enter the human body through ingestion, which can directly endanger human health, causing allergic reactions, disrupting the balance of the human microbiota, affecting the growth and development of children, and even causing cancer [10]. For example, tetracycline antibiotics can inhibit bone marrow hematopoietic function, causing human aplastic anemia [11]; furan antibiotics can cause human hemolytic anemia and acute liver necrosis [12].

In order to ensure the green and sustainable development of aquaculture, it is urgent to find alternatives to antibiotics and chemical drugs. In fact, medicinal plants are very suitable candidates for antibiotic alternatives. Medicinal plants contain a variety of active ingredients, including polysaccharides, alkaloids, organic acids, flavonoids, and phenols, which are antibacterial and antiviral, in addition to promoting the body’s immune function and improving the body’s ability to resist diseases [13]. In recent years, because medicinal plants have the advantages of low toxicity, few side-effects, no drug resistance, few drug residues, and low prices, they have attracted much attention in the prevention and treatment of aquatic animal diseases, and substantial progress has been made in research. For example, methanolic extracts of *Urtica dioica* and *Pleurotus ostreatus* were able to exert antibacterial effects against *Aeromonas hydrophila* in rainbow trout (*Oncorhynchus mykiss*) [14]; *Zingiber officinale Roscoe* enhanced disease resistance of *Lates calcarifer* (Bloch) against *Vibrio harveyi* and enhanced the nonspecific immunity of *Lates calcarifer* (Bloch) [15]; the compound pentagalloylglucose extracted from *Galla chinensis* was highly resistant to *Ichthyophthirius multifiliis* (Ich) and significantly increased the survival of infected channel catfish (*Ictalurus punctatus*) [16]; *Ophiopogon japonicus* extract inhibited the proliferation of white spot syndrome virus (WSSV) in Chinese mitten crab (*Eriocheir sinensis*) and enhanced the immune response of Chinese mitten crab [17].

Viral disease is one of the biggest obstacles facing the aquaculture industry; viruses are highly contagious and fast-spreading, have a wide host range and high mortality rate, and are the most serious type of disease affecting aquaculture species [18]. At present, most viral diseases in aquaculture are not treatable by drugs and are usually preventative in nature. Most farmers can only prevent and control viral diseases via an improvement the aquaculture environment and disinfection, but they do not achieve good results [19]. It is a very urgent task to find more effective methods of virus diseases prevention and control, and herbal medicines have been confirmed by many studies in this regard. Therefore, this article summarizes several common aquatic viruses that are very destructive, collates the components of medicinal plants or natural compounds of plants that have antiviral effects against these viruses, and analyzes the current obstacles and problems that need to be solved in applying medicinal plants to aquaculture, aiming to provide new ideas for the prevention and treatment of viral diseases in aquatic animals.

## 2. Antiviral Effect of Medicinal Plants in Aquaculture

### 2.1. Medicinal Plants against RNA Viruses

#### 2.1.1. Medicinal Plants against IPNV

Infectious pancreatic necrosis virus (IPNV) belongs to the family *Dictyostelium*, a genus of aquatic *Dictyostelium* RNA viruses with icosahedral symmetry, a diameter size of about 60 nm, no capsule membrane, high pathogenicity, and high infectivity [20]. The virus is extremely resistant to the external environment, stable to heat and acid, and insensitive to liposolvents [21]. IPNV mainly infects fry and juveniles up to 20 weeks of age, with a very high mortality rate [22]. Fish diseases caused by IPNV have been observed in Asia, America, and Europe, involving dozens of economic fish and shellfish species [23]. The most obvious pathological feature is pancreatic necrosis with abnormalities in the pancreatic vesicles, islets, and almost all cells, most of which are necrotic [24]. IPNV can be spread both horizontally and vertically; fry infected with IPNV can carry the virus for several years, be excreted in semen, eggs, and feces, and subsequently infect other fish [25]. However, neither the infection mechanism of IPNV nor the immune response of fish is well understood [26].

Species of the genus *Heliotropiu* were capable of inhibiting the replication of viruses such as Herpes simplex virus types 1 and 2, Junin, and Respiratory syncytial virus [27]. Accordingly, Modak et al. [28] evaluated the anti-IPNV effects of a group of natural compounds (geranyl aromatic derivatives) isolated from the resin exudates of *Heliotropium filifolium* (Heliotropiaceae) in vitro, including filifolinol, filifolinone, filifolinyl senecionate, and filifolinoic acid. Geranyl aromatic derivatives were added to the IPNV-infected CHSE-214 cell line at a series of different concentrations, and the viral plaque inhibition was tested. The results showed that the compound ester filifolinyl senecionate presented the best antiviral effect with an EC_50_ of 160 mg/mL and a cytotoxic concentration of up to 400 mg/mL that reduced cell viability by 50%. The experiment indicated that the ester filifolinyl senecionate is a good candidate for the prevention and treatment of IPNV infection in aquaculture.

Mycophenolic acid (MPA) is a non-nucleoside inhibitor of the cellular inosine monophosphate dehydrogenase (IMPDH), an enzyme that catalyzes an essential step in the biosynthesis of guanosine triphosphate (GTP) [29]. The MPA against IPNV and viral hemorrhagic septicemia virus (VHSV) was studied in cell culture [30]. Marroquí et al. evaluated the inhibitory effect of MPA using measurement methods such as RNA and protein synthesis determination, and they found that MPA inhibited the replication of both IPNV and VHSV in a dose-dependent manner. In particular, MPA was particularly effective for IPNV, with the inhibition of infectious virus production more than 10^5^-fold. Additionally, MPA was effective in preventing viral protein synthesis. RT-*q*PCR results showed that MPA showed good efficacy in reducing the accumulation of viral RNA at low concentrations. The time-of-addition and washout experiments suggested that MPA might have a dual mechanism against virus infection that targets both cellular and viral functions. This study demonstrated that MPA could be used as a broad-spectrum antiviral agent for the treatment of aquatic animal diseases.

Infectious pancreatic necrosis (IPN) is a highly contagious acute viral disease that poses a great challenge to salmonid fish farming [31]. IPNV is strongly resistant and can survive for a long time in both fresh and marine water [32]. In the study of this virus, the international vaccine research is progressing well and has reached the molecular level; however, drug research is progressing slowly. There is no efficient drug, and the general research remains at the level of an anti-IPNV effect in cells outside the fish body, whereas the specific practical level in the fish body is inadequate. Moreover, broad-spectrum antiviral drugs have certain toxic effects on cells, and many of them have become prohibited drugs [33]. It is recommended to develop effective control drugs, such as traditional herbal medicines, from a pollution-free perspective.

#### 2.1.2. Medicinal Plants against IHNV

Infectious hematopoietic necrosis virus (IHNV) belongs to the rhabdovirus genus *Novirhabdovirus* [34]. IHNV is a linear single-stranded RNA virus with a genome consisting of unsegmented negative-stranded RNA and a capsule membrane [35]. IHNV is a highly virulent fish bouncing virus that infects rainbow trout, causing acute, fatal systemic disease in fish fry, and renal hematopoietic tissue is the main target organ [36]. Additionally, IHNV is a pathogen that causes severe economic losses in salmonid trout farming in the United States, Canada, and many countries in Asia and Europe [37]. The virus is sensitive to temperature and an acidic environment. However, it is more tolerant in freshwater environments [38]. As with many viral diseases of finfish, younger life stages are more susceptible, with mortality rates of up to 95% during acute outbreaks [39]. Pathological signs in infected fish include protruding eyes, ocular hemorrhage, darkened skin, swollen abdomen, ulceration of the mouth and nose, and yellowish fluid in the intestinal viscera, histopathological examination reveals marked necrosis of the anterior kidney, liver, and spleen [40].

Carotenoids are outstanding antioxidants and have a very important role in the health of both humans and animals. It has been reported that carotenoids can improve the immune function of some aquatic animals [41]. Amar et al. [42] tested changes in resistance to IHNV in rainbow trout after carotenoid administration and found that the survival of rainbow trout did not change significantly when fish were immersed in high concentrations (2 × 10^4^ TCID_50_·mL^−1^) of IHNV; however, when the virus dose was lower (2 × 10^3^ TCID_50_·mL^−1^), the mortality was significantly reduced and the relative survival (RPS) was increased. This study suggested that carotenoids also have a role in enhancing disease resistance in fish.

A novel lentinan (LNT-I), consisting of glucose, mannose, and galactose, was extracted and purified from *Lentinus edodes mycelia* by Ren et al. [43]. LNT-I showed prominent antiviral activity against IHNV at a multiplicity of infectivity (MOI) of 0.05 and 0.10. At an MOI of 0.05, the direct inactivation of LNT-I against IHNV was 62.34% at 100 μg/mL. The antiviral mechanism of LNT-I is mainly involved in the direct inactivation of viral particles and inhibition of viral replication. In addition, LNT-I significantly downregulated the expression levels of tumor necrosis factor-α (TNF-α), interleukin-2 (IL-2), and IL-11, whereas it upregulated the transcription of interferon-1 (*IFN-1)* and *IFN-γ* in epithelioma papulosum cyprini (EPC) cells infected with IHNV, suggesting that the inhibitory effect of LNT-I on IHNV infection might be attributed to its regulation of innate and specific immune responses.

*Prunella vulgaris* L. (PVL) has a broad range of biological activities including immune modulatory, antioxidant, antiviral, anti-inflammatory, antiallergic, and anticancer activities [44]. Li et al. [45] investigated the anti-IHNV activity of 32 medicinal plants in vitro and found that PVL exhibited the strongest antiviral effect with 99.3% inhibition of IHNV at a concentration of 100 mg/L. Further studies showed that ursolic acid (UA), a major component of PVL, also showed efficient anti-IHNV activity. The IC_50_ at 72 h of UA on IHNV was 8.0 μM. In addition, UA significantly reduced the cytopathic effect (CPE) and viral titer in EPC cells infected with IHNV. Moreover, the in vivo experiments also confirmed the strong anti-IHNV capability of UA. When infected with IHNV, the survival rate of the rainbow trout population injected intraperitoneally with UA increased, and viral gene expression was significantly suppressed.

*Rhus verniciflua* Stokes, a deciduous tree of the Anacardiaceae family, has long been used as a food supplement and a traditional herbal remedy for various ailments [46]. The methanolic extract of *Rhus verniciflua* Stokes bark exhibited significant anti-IHNV and VHSV ability and attenuated virus-induced CPE in vitro [47]. Further study of the active substances contained in the extract revealed that fisetin exhibited high antiviral activity against both IHNV and VHSV, as did fustin and sulfuretin. This test showed that *Rhus verniciflua* Stokes bark, as well as the isolated flavonoids, had significant antiviral effects [47]. It is also an excellent antiviral drug candidate.

Infectious hematopoietic necrosis (IHN) is also a viral disease that has dealt a fatal blow to the development of salmon and trout fisheries. To date, the IHNV DNA vaccine has been approved as a commercially available DNA vaccine; however, the lack of a convenient and effective immunization route for the vaccine, the paucity of research on the mechanisms that produce the protective effect of the vaccine, and its impact on the host immune system have limited the widespread use of the IHNV vaccine. In addition, there is no effective treatment for IHNV. According to current research, in addition to the above medicinal plants and natural active ingredients, some active ingredient derivatives are gradually being revealed scientists, such as arctigenin derivatives [48] and coumarin derivatives [49], which are also potential antiviral drugs with very good application prospects that deserve to be studied in depth.

#### 2.1.3. Medicinal Plants against VHSV

VHSV is a member of the genus *Novirhabdovirus*, a family of elasmobranch viruses with a diameter of 65–80 nm, a capsule membrane, and a negative-sense single-stranded RNA genome [50]. The virus was mainly endemic in Europe and North America, but later spread to East Asia [51]. VHSV can be excreted by sick fish and viral feces, eggs, semen, etc. and transmitted through water [52]; it is very stable at low temperatures and can survive longer in freshwater than in seawater [53]. The host range of VHSV is very large, including salmonids, dogfish, cod, and plaice [54]. Fish infected with VHSV have rapid morbidity and mortality rates of 70% to 90% [55]. Infected fish show characteristics of darkening, protruding eyes, swollen abdomen, anemia, hemorrhage on the side of the body and at the base of the fins, hemorrhage, swelling, and discoloration of the liver, kidneys, and other organs, and punctate hemorrhage in the skeletal muscle. When a pathological histological examination was performed, necrosis of the urinary system and hematopoietic tissues such as the kidneys, some necrosis of the liver and spleen, and marked hemorrhage in the skeletal muscles were seen [56].

*Celosia cristata* and *Raphanus sativus* are herbs frequently used in traditional medicine. Park et al. [57] tested the antiviral activity of a mixed extract of *Celosia cristata* and *Raphanus sativus* against VHSV, and they found that, if the extract was added before the EPC cells were inoculated with the virus, it had a good effect in reducing the virus titer. However, the addition of this extract after the cells were inoculated with the virus had no effect. In an in vivo assay, the expression of several immune-related factors was examined in fish orally administered with the extract. The results showed that the expression of Mx and TLR7 peaked at 48 h after extract administration, while the expression of ISG15 and TLR2 peaked at 72 h, indicating that the antiviral activity of *Celosia cristata* and *Raphanus sativus* is achieved by inducing the expression of genes involved in the innate immune response.

Licorice (*Glycyrrhiza uralensis*, GUF) is a very common medicinal plant, and many important research projects have identified the many beneficial effects of this medicinal herb, including antimicrobial, antiviral, anti-inflammatory, antioxidant, antiprotozoal, hepatoprotective, and neuroprotective activities. Licorice contains important bioactive components, such as glycyrrhizin (GL), glycyrrhizinic acid (GLA), licorice acid, formononetin, and liquiritin [58]. Licorice extracts showed significant antiviral activity against VHSV, significantly reducing virus-induced CPE [59]. The active pharmaceutical ingredients GL and GLA also showed antiviral activity against VHSV. Moreover, a time-course study using a plaque reduction assay showed that GLA exhibited direct virucidal effects against VHSV or might inhibit viral invasion into cells.

The antiviral effects of curcumin have been extensively studied, and the mechanism of action mainly involves direct interference with the viral replication machinery, as well as inhibition of cellular signaling pathways essential for viral replication, such as phosphatidylinositol-3-kinase (PI3K)/Akt and nuclear factor kappa B (NF-κB) [60]. Curcumin also has an inhibitory effect on VHSV [61]. Pretreatment with 120 mM curcumin resulted in a significant increase in the viability of VHSV-infected cells, a decrease in viral copy number, and a decrease in apoptosis. A further study found that curcumin inhibited VHSV entry into cells by downregulating fibronectin (FN) 1 or upregulating F-actin. HSC71 is the primary protein interacting with FN1, gelsolin (GSN), and actins, which is a key target for protection from VHSV infection at the viral entry stage. Indeed, curcumin downregulated HSC71 expression while increasing the viability of cells infected with VHSV and inhibiting VHSV replication. The present experiments suggest that curcumin antiviral might be achieved by downregulating the expression of HSC71.

Olive (*Olea europaea*) (LExt) is the most famous member of the genus *Olea*, has a high nutritional value, and is commonly used by people; extracts from its leaves and fruits are widely used to counteract and prevent various pathologies [62]. It contains a significant number of bioactive compounds such as flavonoids, secoiridoids, carotenoids, and phenolic compounds. Olive tree leaf and its active component oleuropein (Ole) were able to inhibit VHSV infection in vitro, and incubation of the virus with LExt or Ole prior to infection reduced viral infectivity to 10% and 30%, respectively [63]. Furthermore, when LExt was added for 36 h after viral infection of cells, LExt sharply reduced VHSV titer and viral protein accumulation in a dose-dependent manner. On the other hand, both LExt and Ole were able to inhibit VHSV transmission between cells. This demonstrates that *O. Europaea* is a promising source of natural antiviral drugs.

α-Lipoic acid (LA) is a disulfide-containing compound that acts as a cofactor for several enzymes participating dehydrogenation and decarboxylation. Due to its excellent antioxidant capacity, LA has been used as an aquafeed additive to enhance detoxification and antioxidant capacity [64]. Furthermore, LA significantly increased the antiviral ability of fathead minnow cells against VHSV in a dose-dependent manner [65]. The dosing time assay showed that the antiviral activity of LA occurred during the viral replication phase. Injecting LA and VHSV into *Micropterus salmoides* also significantly enhanced survival compared to VHSV-only controls. In addition, LA injection significantly upregulated the expression of several immune-related genes, including IRF7, Viperin, and ISG15, as well as reduced VHSV-induced oxidative stress. (Figure 1).

VHSV has been found in most countries in the Northern Hemisphere, and VHSV is frequently endemic in aquaculture, causing morbidity and mortality in a wide range of fish species. Both VHSV and IHNV are members of the *Novirhabdovirus* family and are biologically similar in that they are both highly resistant to the external environment. After an epidemic, survivors are usually in a carrier state, and these carriers of the virus spread the pathogen throughout their lives via urine, feces, or gills. There is no commercially available vaccine for VHSV or effective prevention and control measures. In the study of this virus, more research has been conducted on diagnosis and detection, but less research has been invested in the study of prevention and control drugs; thus, we should pay more attention to the development of new efficient and nonpolluting drugs against VHSV.

#### 2.1.4. Medicinal Plants against SVCV

Spring viremia of carp virus (SVCV) belongs to the family *Rhabdoviridae*, genus *Vesiculovirus*, and it is a negative-sense single-stranded RNA virus with a capsule membrane [66]. SVCV can infect koi (*C. carpio koi*), rainbow trout, sockeye salmon (*O. nerka*), blackhead minnow (*Pimephales promelas*), yellow perch (*Perca flavescens*) [67], and common carp (*Cyprinus carpio carp*), among them, common carp is the main host. SVCV is a serious pathogen of several economically important carp species, with a lethality rate of up to 100% [68]. SVCV was first identified in Europe and has since been found in Asia and South America [69]; it has spread widely throughout carp culture mainly in Europe and has caused significant economic losses [70]. The clinical symptoms of the diseased fish are mainly slow movement, neurological disorders, internal bleeding, abdominal inflammation, and fluid accumulation in the abdomen [71]. SVCV can be transmitted through the water by mucus-like feces excreted by diseased fish and also by blood-sucking insects, and it invades into fish via gills or other routes [72]; it can also replicate in fish epithelial cells [73]. SVCV is mainly transmitted horizontally, while it can also be transmitted vertically [74].

Beta-glucans (β-glucans) are glucose polymers found in yeast cell walls. There have been many literature studies on the expression of immune response genes in fish, where a β-glucan-enriched diet increases the transcript levels of some key genes of the innate and adaptive immune response [75]. β-Glucans have a therapeutic effect on zebrafish infected with SVCV via a mechanism that mainly regulates the release process of inflammatory factors [76]. In vitro tests revealed that β-glucose was able to regulate the location of IL-1 in cells; in vivo tests revealed that the expression of a group of genes involved in the innate immune response (IL-1β, IL-6, IL-8, IL-10, and TNF-α) were upregulated to inhibit the process of SVCV infection.

Tian et al. [77] purified the chemical element selenium from the fermentation of *Herbaspirillum camelliae* WT00C. Selenium is an essential trace element with antiviral properties. Selenium has various functions, such as antioxidant, anti-inflammatory, and immune boosting activities, in addition to reducing the incidence of cancer [78]. Tian and his subject group found that selenium significantly induced the expression of IFN, ISG15, and Mx, key factors in the immune response process. In in vivo experiments, IFN expression was increased 13- and 39-fold in *Carassius carassius* fed food containing 5 and 10 μg/g elemental selenium, respectively. On the 16th day post injection, the expression of IFN was selenium concentration-dependent within a certain range. The antiviral effects of selenium are mainly derived from its immunomodulatory effect through its incorporation into selenoproteins. The right amount of selenium helps to improve the immunity of fish, while excess selenium causes excessive immune and inflammatory responses.

*Astragalus membranaceus* is a traditional herb with a long history of clinical application. *Astragalus* polysaccharide (APS) is an important bioactive component of *A. membranaceus* and has a variety of pharmacological properties [79]. The antiviral effect of APS in zebrafish was investigated by Li et al. [80]. They fed zebrafish for 3 weeks on the control diet and experimental diets containing 0.01% or 0.02% APS. Zebrafish were challenged by SVCV at the end of a feeding. The antiviral immune responses were assessed 4 days after SVCV infection, and the survival rate was calculated 14 days after the challenge. The results showed that zebrafish fed 0.01% APS exhibited a higher survival rate post SVCV infection. Correspondingly, the expression of antiviral genes in the spleen was increased 4 days after the challenge. However, the addition of excessive amounts of APS was detrimental to liver health.

Librán-Pérez et al. [81] studied the effect of palmitic acid (PA) treatment on different immune parameters and mortality caused by SVCV in zebrafish. The results showed that PA had a positive effect on the immunomodulation of zebrafish via reducing mortality and viral titers. This FA-induced antiviral protective effect appears to be associated with the inhibition of autophagy and be independent of other immune processes such as neutrophil proliferation or IFN activity. PA is the most common saturated fatty acid and has long been negatively portrayed for its adverse health effects, but it actually has a variety of important physiological activities [82]. The usage of PA as an immunostimulant at low concentrations has shown great potential in the prevention of SVCV infection.

*Psoralea corylifolia* Linn. is a proverbial traditional medicinal plant that has been used since ancient times for the treatment of various diseases. It is widely distributed and is an important part of therapeutics in Ayurveda and Chinese medicine [83]. *Psoralea corylifolia* Linn. and one of its main components, bavachin (BVN), inhibited SVCV replication strongly [84]. BVN significantly inhibited both SVCV glycoprotein and nucleoprotein expression, but BVN did not affect the infectivity of SVCV and could not be used to prevent SVCV infection. Mechanistically speaking, BVN mainly inhibited the early events of SVCV replication but did not interfere with SVCV adsorption, and the inhibition of SVCV replication by BVN was achieved to some extent by blocking SVCV-induced apoptosis. This trial demonstrated that BVN could inhibit SVCV replication and block viral release to protect host cells, leading to therapeutic effects.

Saikosaponin D (SSD), a major bioactive triterpenoid saponin, an important component of *Bupleurum yinchowense*, significantly reduced SVCV-induced apoptosis and recovered SVCV-activated caspase-3/8/9 activity [85]. Moreover, SVCV-induced cell morphological damage and reactive oxygen species (ROS) generation were inhibited by SSD treatment. In vivo tests showed that SSD was also effective in the treatment of SVCV-infected zebrafish and carp. Intraperitoneal injection of SSD (6 mg/kg) increased the survival rate of zebrafish up to 36% and inhibited the expression of more than 90% of SVCV nucleoprotein and glycoprotein genes in the kidney and spleen, while the survival rate of carp treated with SSD (6 mg/kg) was also found to increase by 32%. In conclusion, SSD has been proven to be a natural product with very high anti-SVCV activity.

Spring viremia of carp (SVC) is one of the most damaging infectious diseases in the carp family, with diverse modes of transmission, long in vitro survival time, and high infective activity of virus particles expelled from infected fish. Although substantial research has been conducted worldwide on the control of SVC, the results have not been fruitful. There are no recognized drugs, vaccines, or other methods for effective control of the disease, and the internationally accepted prevention and control measures are to closely monitor SVC, detect diseased fish early, and isolate and cull them, so as to interrupt the trend of the disease outbreak and epidemic. Considering the sustainability of the entire carp culture industry, the prevention of the disease at its root also requires the use of resourceful and inexpensive herbal medicine containing a large number of effective natural components and various secondary metabolites.

#### 2.1.5. Medicinal Plants against NNV

Nervous necrosis virus (NNV) is a widespread and serious infectious agent with a spherical shape, non-capsule membrane, diameter of approximately 30–40 nm, and a single positive-stranded, 2-segmented RNA molecule genome [86]. NNV belongs to the family of *Nodaviridae*, β genus Noda virus [87], which is divided into four genotypes (SJNNV, RGNNV, TPNNV, and BFNNV) [88]. The optimum temperature of NNV is 20–30 °C; NNV infections are usually observed in aquaculture farms when seawater temperatures are above 24 °C [89]. NNV usually infects fish fry and causes viral neuronecrosis, encephalomyelitis, and vacuolar retinopathy in fish after infection, resulting in massive fish mortality, which poses a devastating threat to aquaculture worldwide [90]. However, the pathogenic mechanism is still poorly understood [91]. Moreover, there is no effective vaccine to prevent the disease [92].

*Gymnema sylvestre* is mainly found in tropical and subtropical regions and has unique therapeutic properties for diabetes [93]. The gymnemagenol (C_30_H_50_O_4_) extracted from the leaves of *Gymnema Sylvestre* had antiviral potential [94]. In in vitro conditions, they used Sahul Indian Grouper Eye (SIGE) cell lines infected with grouper nervous necrosis virus (GNNV) to study the antiviral activity of gymnemagenol. The inhibition of GNNV by gymnemagenol was confirmed by measuring the viral titer (TCID_50_·mL^−1^) in the infected SIGE cells every 24 h. It was found that gymnemagenol exhibited a concentration-dependent inhibition on GNNV proliferation in virus-infected SIGE cells. At the end of the sixth day, 20 μg·mL^−1^ of gymnemagenol inhibited the proliferation of GNNV to 53%, while at the end of day 7, 20 μg·mL^−1^ of gymnemagenol reduced the survival rate of virus-infected SIGE cells to 47%. We conclude that gymnemagenol can be used as an antiviral agent against GNNV infection.

In addition, Asim et al. [95] studied the effect of glutamine on RGNNV replication. They found that a lack of glutamine did not affect cell viability but greatly inhibited RGNNV replication, revealing that glutamine is required for RGNNV replication. Glutamine can be converted to α-ketoglutarate (α-KG) by glutaminase (GLS) and then incorporated into the tricarboxylic acid (TCA) cycle. Inhibition of GLS activity with a GLS inhibitor significantly inhibits RGNNV replication while increasing the intermediates of the TCA cycle. The addition of α-KG, oxaloacetic acid (OAA), or pyruvate to glutamine-free medium significantly restored RGNNV replication, indicating that glutamine is required for RGNNV proliferation to complement the TCA cycle. These data suggest that glutamine could regulate RGNNV replication through the TCA cycle, which would pave a new pathway for the prevention of RGNNV infection in the future [95].

As mentioned earlier, β-glucan was able to increase the expression of some genes critical for immune response. Krishnan et al. [96] investigated the anti-NNV effect of β-glucan and found that β-glucan enhanced the viability of macrophages against NNV, which was associated with the activation of gene expression of persistent inflammatory cytokines. In the experiment, the expression of both NLR family CARD domain-containing 3 (NLRC3) and caspase-1 in the test group of seven-banded grouper was higher in the early stage and lower in the late stage than in the control group, suggesting that β-glucan may induce an antiviral state against NNV infection in grouper macrophages. In the present experiment, macrophages attenuated viral infectivity without cell death, as the lactate dehydrogenase (LDH) center remained stable in both the test and the control groups. In conclusion, it is known from the present experiment that β-glucan induces a protective response of the host against NNV infection.

Viral encephalopathy and retinopathy (VER) and viral nervous necrosis (VNN), caused by NNV, have no effective control measures, whether drugs or vaccines. Only a few herbal medicines have entered the anti-NNV research field in recent years, but there is still a lack of direct antiviral evidence, and the mechanism of action is still unclear; the lack of research in natural drug treatment has resulted in a serious shortage of natural drugs in the antiviral field. NNV has a very wide range of infection, including numerous marine invertebrates. The mortality rate of fish fry in farms infected with NNV is extremely high. Therefore, it is also extremely important to search for medicinal plants with high antiviral activity against NNV and to conduct systematic studies.

#### 2.1.6. Medicinal Plants against GCRV

The grass carp reovirus (GCRV), a member of the *Reoviridae* family, *Aquareovirus*, is a double-stranded RNA virus with spherical particles, approximately 75 nm in diameter, which possesses a double non-enveloped capsid [97]. GCRV particles are insensitive to lipid solvents and are stable between pH 3 and 10. Moreover, these viruses are heat-stable and remain infectious after 30 min of passivation at 56 °C [98]. GCRV is the most pathogenic strain of the genus *Echovirus* [99] and can infect not only grass carp, but also cyprinid fish, whelk, and buccaneer [100]. The typical symptoms of GCRV infection are muscle and visceral hemorrhage, black body color, and unresponsiveness [101]. GCRV enters the fish mainly through the gills. Fish infected with GCRV produce a symptom reaction 4 days after infection. After 5–7 days, the fish show obvious hemorrhagic symptoms, and the fish population enters the peak of mortality [102]. Because of its high risk, wide prevalence, and long onset season, this virus has been the focus of research on fish viruses, but there is not yet an effective treatment [103].

Chen et al. [104] conducted in vitro experiments using *Ctenopharyngodon idella* kidney (CIK) cells to investigate the effects of 30 plant extracts on GCRV viral mRNA expression, and they found that *Magnolia* bark extract exhibited very strong inhibition of GCRV replication in vitro. Further studies showed that the main components of *Magnolia* bark, namely, magnolo and honokiol, were both effective in inhibiting GCRV replication in CIK cells at safe concentrations, but the mechanisms were not the same. In GCRV-infected cells, magnolol induced IFN-I activation by enhancing IRF7 expression, while honokiol enhanced the innate antiviral response of the host to GCRV infection via the NF-κB pathway [105]. They conducted a further study on magnolol and found that magnolol increased the resistance of grass carp to GCRV infection and inhibited GCRV-induced apoptosis; the inhibition of this apoptotic effect was probably due to the direct interaction of magnolol with caspase-3 [104].

Epigallocatechin-3-gallate (EGCG) is the most abundant and active compound in green tea. EGCG has significant antimicrobial, anti-inflammatory, antioxidant, anticancer, immunomodulatory, and neuroprotective effects, and has therapeutic potential for many diseases [106]. The adhesion of GCRV particles to the surface of CIK cells was inhibited by EGCG, as well as crude extracts of green tea, in a dose-dependent manner [107]. Through the virus overlay protein binding assay (VOPBA), they deduced that the blocking effect of EGCG on GCRV adhesion was due to EGCG serving as a ligand for the 37/67 kDa laminin receptor (LamR), which was related to the binding potential of GCRV particles to LamR [107]. Further studies showed that the addition of a certain dose of EGCG to fish feed could improve the survival rate of grass carp. This experiment also showed that even high doses of EGCG did not cause toxicity, and there was no significant change in cell survival at high concentrations. Grass carp injected with EGCG showed a decrease in malondialdehyde (MDA) concentration and an increase in glucuronide (GSH) and lysozyme (LZM) concentrations compared to the control group [108].

Zhang et al. [109] evaluated the effect of epicatechin-3-gallate (ECG) on GCRV in vitro, which differs from EGCG only in the absence of hydroxyl groups. ECG and EGCG have similar pharmacokinetic behavior, and EGCG is partially converted to ECG in vivo. GCRV was incubated with different concentrations of ECG in CIK cells and it was found that the typical CPE was mild in the range of 20–40 μg/mL ECG, while too high or too low concentrations caused harm to CIK cells. Western blot results showed that ECG efficiently inhibited the replication of GCRV in CIK cells. These articles suggested that both ECG and EGCG exhibit potential as antiviral agents in aquaculture.

Ginsenosides are the compounds responsible for the main pharmacological effects of *Panax ginseng*. Dai et al. [110] investigated the inhibitory effect of ginsenoside Rg3 on GCRV-infected grass carp ovarian (CO) epithelial cells. This experiment treated CO epithelial cells with 1, 10, and 100 μg/mL ginsenoside Rg3 and found that the highest inhibitory effect on GCRV was observed with 100 μg/mL ginsenoside Rg3 treatment. The antioxidant assay and RT-*q*PCR assay showed that ginsenoside Rg3 could effectively inhibit the replication of GCRV in CO epithelial cells. Expression analysis of immune-related genes showed that treatment with ginsenoside Rg3 promoted the expression of IRF-3 and IRF-7, induced the expression of IFN-1, and inhibited the expression of TNF-α. In conclusion, this experiment demonstrated that ginsenoside Rg3 enhanced the resistance of CO cells to GCRV infection by inhibiting GCRV activity and promoting the immune activity of CO cells.

Previous studies have shown that heat-shock factor 1 (HSF-1) is responsible for the transcriptional activation of heat-shock proteins in mammals in response to stress stimulation, and quercetin effectively inhibits HSF-1 transcription [111]. Shan et al. [112] found that heat-shock protein 70 (Hsp70), the major heat-shock protein during the heat-shock response, played an important role in facilitating the entry of GCRV into CIK cells and acted as a pro-viral factor during GCRV infection. These results suggested that quercetin (Qct) is likely to have anti-GCRV effects. Thus, Fu et al. [113] designed a relevant experiment and confirmed that quercetin could effectively improve the survival rate of the rare minnow *Gobiocypris rarus* after GCRV infection. Qct was injected at a dose of 30 µL/tail at concentrations of 1, 2, 4, 6, 8, and 16 mg/mL, and the mortality of rare minnow at different concentrations of Qct was observed and recorded over a period of 15 days; the results showed that quercetin dose-dependently protected rare minnow from GCRV challenge by reducing the production of progeny virus and improving fish survival. Hence, quercetin should be used as a promising environmentally friendly therapeutic agent for aquaculture.

Grass carp hemorrhagic disease (GCHD) is characterized by a wide range of prevalence, a long incidence season, a high incidence rate, and easy mutation of the pathogen. At present, although the research on vaccine control of GCHD has made great progress, the frequent mutation of GCRV and the bottleneck of vaccine commercialization for large-scale application have restricted the process of research and application of disease control, whereas the use of drugs for control is still the most direct and effective method for large-scale control of grass carp hemorrhagic disease. As a result, the development of antiviral drugs and related mechanisms of research are particularly important in the field of grass carp hemorrhagic disease control. The current research on the drug control of GCRV is still extremely backward, and the pharmacology of herbal drugs is not clear; thus, aquarists need to accelerate the pace of research.

### 2.2. Medicinal Plants against DNA Viruses

#### 2.2.1. Medicinal Plants against SGIV

The grouper iridovirus (SGIV) is a novel marine fish DNA virus that belongs to the family *Iridoviridae*, genus *Ranavirus* [114]. SGIV is a class of plasma type linear double-stranded DNA viruses with a capsule membrane, virions in the form of regular icosahedrons, and a diameter of 154–176 nm. It is highly contagious; grouper fry are susceptible to this disease, with a large number of deaths in a short period of time after the disease [115]. SGIV is highly pathogenic, and high morbidity and mortality rates in aquatic animals have been caused by iridovirus infections worldwide in recent years [116]. The disease caused by SGIV results in significant economic losses to the aquaculture industry and poses a major threat to global biodiversity. Accordingly, SGIV has attracted the attention of virologists [117]. The main manifestation symptoms of iridovirus disease are fish swimming alone, white or bleeding spots on the gill filaments, swollen and white or earthy yellow liver, and enlarged liver and spleen [118]. An inactivated SGIV vaccine is currently available on the market, and the protective effect produced by the vaccine can be observed within 30 days of intraperitoneal injection to grouper [119]. However, inactivated whole virus vaccines have certain drawbacks, such as the risk of incomplete inactivation and the higher dose and cost of immunization in aquatic animals compared to terrestrial animals [120]. More importantly, vaccines cannot treat viral diseases; thus, it is imperative to develop new efficient and safe antiviral drugs to control the SGIV epidemic in aquaculture.

*Illicium verum* Hook. f. is rich in chemical constituents, and has antiviral, antibacterial, and antiphlogistic properties [121]. Liu et al. [122] investigated the application of *I. verum* extracts including *trans*-anethole (TAT), 3,4-dihydroxybenzoic acid (DDBA), and Qct in the treatment of grouper iridovirus infection, and the highest safe concentration of the extracts was selected for subsequent tests. The inhibitory activity of each *I. verum* extract against grouper iridovirus infection was analyzed using an aptamer (Q2)-based fluorescent molecular probe (Q2-AFMP) and the RT-*q*PCR method. All *I. verum* extracts showed dose-dependent antiviral activity against grouper iridovirus. IVAE, IVEE, DDBA, and Qct all showed maximum antiviral activity (all >90%) according to the percentage inhibition achieved.

Later, they further analyzed the possible antiviral mechanism of Qct [123,124]. Qct resulted in significant damage to SGIV particles. Furthermore, Qct could interfere with the binding of SGIV to host cell targets (76.14%), prevent SGIV invasion into host cells (56.03%), and affect SGIV replication in host cells (52.73%). The *q*PCR assay revealed that Qct induced the expression of genes for immune-related factors (IFN, STAT1, PKR, MxI, and ISG15) and TLR9. This suggested that quercetin exerted indirect antiviral activity against SGIV infection by promoting SGIV recognition and activating the IFN pathway to establish the antiviral state of host cells. Overall, Qct has direct and host-mediated antiviral effects against SGIV and has great potential for both prevention and treatment of SGIV infection.

In addition to the *I. verum* extracts, some active pharmaceutical ingredients of *Curcuma kwangsiensis* also exhibited strong anti-SGIV effects. *C. kwangsiensis* belongs to the family Zingiberaceae, which is used in medicine and food for the improvement and treatment of various diseases [125]. Liu et al. [126] tested *C. kwangsiensis* ethanol ingredient (CKEE), curcumin, curdione, curcumenol, and curcumol in vitro and in vivo. The results of the aptamer Q2-AFMP and RT-*q*PCR showed that all of them exhibited antiviral activity in a dose-dependent manner. CKEE and curdione showed the best results with inhibition rates higher than 93%, and they are excellent candidates for the development of effective drugs for the prevention and control of SGIV outbreaks in mariculture.

Liu et al. [127] found that *Lonicera japonica* Thunb. extracts also showed anti-SGIV activity, providing new ideas for the development of effective drugs for the prevention and control of SGIV infection in mariculture. They examined *L. japonica* aqueous extracts (LAE) and the active pharmaceutical ingredients chlorogenic acid (CGA), cryptochlorogenic acid (CCGA), isochlorogenic acid A (IAA), isochlorogenic acid B (IAB), isochlorogenic acid C (IAC), caffeic acid (CA), luteolin (LT), and inositol (IS) by light microscopy, Q3-AFMP, and RT-*q*PCR, revealing their anti-SGIV effects, with an inhibition rate above 90% except for IS.

*Viola philippica* is a perennial herb distributed throughout East Asia, containing anthocyanins, flavonoids, coumarins, alkaloids, phytosterol, sulfonated carbohydrate polymer, cyclotides, etc. [128]. The aqueous extracts of *V. philippica* also showed strong inhibition of SGIV infection both in vitro and in vivo [129]. RT-*q*PCR results showed that *V. philippica* did not damage SGIV particles, but it could interfere with SGIV binding, entry, and replication in host cells. *V. philippica* was found to have the best inhibitory effect on SGIV; it was very effective at the stage of virus binding and replication. In conclusion, the results indicate that the aqueous extract of *V. philippica* at appropriate concentrations has a strong anti-SGIV effect.

As previously mentioned, *Glycyrrhiza uralensis* (GUF) has a variety of beneficial effects, including but not limited to antiviral, antibacterial, and anti-inflammatory activities. Li et al. [130] investigated the antiviral effects of licorice against SGIV and found several major components of GUF, glycyrrhizin, glycyrrhetinic acid, liquiritin, isoliquiritigenin, and liquiritigenin. Although none of them exhibited significant anti-SGIV effects, the aqueous extract of GUF had significant anti-SGIV infection activity in a concentration-dependent manner. Moreover, the aqueous extract of licorice could disrupt the particle structure of SGIV. These results suggest that its antiviral effect may occur during the binding of viral particles to cellular receptors and during the replication phase of the virus in host cells, but the exact components of its action are not known and need to be further investigated.

#### 2.2.2. Medicinal Plants against WSSV

WSSV is a rod-shaped double-stranded DNA virus containing a capsid membrane, approximately 250–300 nm × 75–100 nm in size, belonging to the family *Nimaviridae*, genus *Whispovirus* [131]. WSSV is highly infectious and lethal. It is reported that WSSV mainly infects crustacean decapods such as *Penaeus monodon*, *P. vannamei*, *P. japonics*, *P. chinensis*, *P. merguiensis*, *P. penicillatius*, and *Scylla paramosain* [132]. Under natural conditions, WSSV can be transmitted both horizontally through excreta, diseased shrimp remains, etc., by feeding and contacting with body parts, and vertically by infecting offspring individuals [133]. WSSV is very destructive; the cumulative mortality of infected shrimp can reach 100% within 3 to 10 days [134]. Shrimp infected with WSSV show symptoms such as lethargy, empty stomach, floating head, loss of balance, and white spots on the carapace, especially on the cephalothorax. In the final stage of the disease occur cell disintegration and organ necrosis, and eventual disease shrimp death [135]. To date, WSSV remains the number one killer of shrimp culture worldwide. At present, the occurrence of white spot syndrome is a multifactorial effect that is closely related to environmental conditions, culture density, etc. [136]. In contrast to the natural environment, shrimp culture in farms with high densities and unstable water conditions can easily form outbreaks once WSSV infection occurs, making it difficult to control [137].

EGCG has anti-WSSV activity in both *Scylla paramosain* [138] and *Marsupeneaus japonicus* [139], and the resistance of *S. paramosain* and *M. japonicus* to WSSV was significantly increased after EGCG treatment. Further studies revealed that moderate amounts of EGCG could effectively enhance innate immunity in shrimp and crab, and that EGCG had positive effects on several innate immune-related genes, including the immune deficiency pathway (IMD), prophenoloxidase (proPO), QM, myosin, Rho, Rab7, p53, TNF-α, mitogen-activated protein kinase (MAPK), and nitric oxide synthase (NOS). EGCG may induce certain immune pathways in vivo, such as the Janus kinase (JAK)–signal transducers and activators of transcription (STAT) pathway. EGCG also enhances phenol oxidase and superoxide dismutase activities and is an excellent anti-WSSV drug candidate [138,139].

Sun et al. [140] compared the inhibitory activity of 13 herbs against WSSV and found that the extract of *Typha angustifolia* showed a very strong inhibition of WSSV replication (84.62%, 100 mg/kg). Further studies showed that its main active ingredient, naringenin (NAR), a citrus flavonoid that possesses various biological activities, exhibited higher inhibition of WSSV (92.85%, 50 mg/kg). NAR inhibited the transcription of key genes during viral replication, especially the immediate-early gene ie1. NAR blocked the transcription of ie1 by reducing the expression of STAT gene and regulated several anti-inflammatory-, antioxidant-, and proapoptotic-related factors, such as cyclooxygenase (COX)-1/2, catalase (CAT), and Bax, to inhibit WSSV replication. These results suggest that NAR could be a candidate for the prevention and treatment of WSSV [140].

Hesperidin, an important constituent of citrus fruits, has a wide range of pharmacological effects, including antiviral, anti-inflammatory, antioxidant, antitumor, and radioprotective activities [141]. The addition of 50–150 mg·kg^−1^ hesperidin (the major compound of Pericarpium Citri Reticulatae) to diets fed to *Procambarus clarkii* not only reduced the percentage of mortality following WSSV infection in *P. clarkii*, but also improved their nonspecific immunity, antioxidant capacity, and growth performance [142]. Compared with the control group, the crayfish group supplemented with 50–150 mg·kg^−1^ hesperidin showed increased final body weight (FBW), specific growth rate (SGR), and weight gain (WG), as well as significantly increased total antioxidant capacity (T-AOC), glutathione peroxidase (GPx) activity, and superoxide dismutase (SOD) activity. Moreover, the activities of acid phosphatase (ACP), alkaline phosphatase (AKP), LZM, and phenoloxidase (PO) were increased in crayfish, and the mRNA expression of extracellular copper-zinc superoxide dismutase (ecCuZnSOD), Hsp70, cyclophilin A (CypA), GPxs, astacidin, crustin, and Toll 3 was upregulated.

Huang et al. [143] evaluated the anti-WSSV activity of 23 medicinal plant extracts and found that *Gardenia jasminoides* extracts showed the highest inhibition of WSSV replication (92.31%). After further study, they found that *G. jasminoides* reduced the transcription of the immediate-early gene ie1, DNA pol, and the late gene VP28 of WSSV, while inhibiting the expression of Hsp70. In addition, after injecting *G. jasminoides* into *P. clarkii*, the expression of antioxidant- and apoptosis-related factors, such as cMnSOD, mMnSOD, and Bax, increased significantly, indicating that *G. jasminoides* not only has antiviral but also antioxidant activity, in addition to regulating apoptosis-related factors. Next, they extracted the active compound genipin (GN) from the fruits of *G. jasminoides* and investigated the anti-WSSV effect of GN; they found that GN decreased the expression of STAT gene, thus reducing the transcription of the ie1 gene. In addition, GN upregulated the expression of antioxidant-related genes and downregulated the expression of inflammation-related genes, thereby reducing oxidative stress and inflammation caused by viral infection.

Huang et al. [144] also found that the extract of *Eucommia ulmoides* showed very high inhibition of WSSV replication (84.12%); its main active ingredient, geniposidic acid (GPA), exerted a very strong antiviral effect (over 97%). Moreover, when WSSV was reinfected with *P. clarkii* after treatment with GPA, the mortality rate of *P. clarkii* was reduced by 43.33%. GPA is an iridoid glucoside, a major component of several herbs that has long been used to treat inflammation, jaundice, and hepatic disorders [145]. Mechanistically, GPA reduces the transcription of ie1 gene by decreasing the expression of STAT, thus inhibiting the replication of WSSV. In addition, GPA also inhibited the expression of the apoptosis-related factor Bax inhibitor-1, which also indirectly inhibited WSSV replication. Pursuantly, GPA can also be considered as a potential drug for the prevention and treatment of WSSV infection.

There are many more candidates like this, such as *Pongamia pinnata* [146], *Ulva intestinalis* [147], and *Gracilaria tenuistipitata* [148]. WSSV is one of the most thoroughly studied aquatic viruses to date, as the major disaster it has brought to shrimp and crabs is one of the problems that fisheries and biologists still have to face.

WSSV is one of the most serious pathogens in shrimp aquaculture, causing catastrophic economic losses to the global shrimp industry and seriously hindering the sustainable development of the marine economy. Many scholars have conducted substantial research on the epidemiological characteristics, etiology, pathology, pathogenesis, and rapid detection techniques of WSSV, and considerable progress has been made. However, due to the habits of shrimp themselves and the complexity of WSSV transmission, the mechanism of WSSV transmission is not yet fully understood, resulting in the slow development of its prevention and control; furthermore, the existing technology has not been able to fully and effectively control the WSSV epidemic. Shrimp lack a vertebrate-like adaptive immune system, but an effective innate immune system may render protection against invading pathogens [149]. An effective way to improve the resistance of shrimp to pathogenic invasion is by enhancing nonspecific immunity. Medicinal plants and their natural chemical components are very effective in this regard, activating the body’s immune system and enhancing the resistance of shrimp to disease. Thus, the use of medicinal plants against WSSV is an excellent strategy.

Table 1 summarizes the medicinal plants and active pharmaceutical ingredients that can inhibit the abovementioned viruses, as well as their effects or possible antiviral mechanisms. It should be noted that the antiviral mechanisms of some of these medicinal plants and active pharmaceutical ingredients have not been confirmed, but are merely inferences made by researchers on the basis of past results.

## 3. Challenges and Perspectives

The rise of aquaculture is considered to be one of the most profound changes in global food production in a century [178] and one of the most environmentally friendly and sustainable food industries to meet the needs of humans today [179]. However, almost all kinds of farmed aquatic animals are threatened by infections from viruses, bacteria, parasites, or other nascent and regenerating pathogenic microorganisms [180]. Epidemics have become a constraint to the sustainable development of the aquaculture industry, especially viral diseases, which are highly contagious, spread rapidly, and have a wide range of hosts with high mortality rates [181]. The occurrence of aquatic animal viral diseases not only leads to a decline in the quality of aquatic products and food safety hazards, but also seriously hinders the sustainable development of fisheries.

Medicinal plants and active pharmaceutical ingredients have unique advantages in terms of antiviral activity. On the premise of being able to effectively inhibit viral infection, medicinal plants and active pharmaceutical ingredients possess the advantages of low drug resistance, fewer toxic side-effects, fewer drug residues, and less pollution of the farmed water environment [182]. Consequently, the search for new antiviral drugs with high efficiency and low toxicity from natural plants is an important way to develop antiviral drugs. At present, the development and the research of safe and efficient new herbal medicine products are getting more and more attention. An investigation found that healthy aquaculture has certain advantages and great potential for the future [183]. The antivirus mechanisms of medicinal plants are diverse, such as direct inactivation of viral particles [184], interdiction of viral attachment and penetration phases [185], inhibition of virus replication [186], involvement in transcriptional regulation [187], disruption of virus protein synthesis or expression [188], inhibition of viral cell-to-cell transmission [189], and immunomodulatory roles [190]. The same medicinal plants may exert different mechanisms of action against different viruses, and different medicinal plants exert even more different effects, which makes the utilization of medicinal plants promising.

Medicinal plants contain numerous active ingredients such as phenolic substances, flavonoids, alkaloids, terpenoids, pigments, starch, steroids, and essential oils [191]. In aquaculture, medicinal plants and active ingredients can be used as growth promoters [192], immunostimulants [193], antibacterial agents [194], antifungal agents [195], antistress agents [196], appetite stimulants [197], and even aphrodisiacs [198], in addition to being antiviral agents. In many cases, medicinal plants exert antiviral effects by acting as immunostimulants, with antiviral activity being only an added effect. At this point, although researchers have used particular viruses to confirm whether a medicinal plant exerts an antiviral effect in fish or shellfish, it is likely that the resistance effect is not limited to that virus. When host nonspecific immunity is increased, it is usually elevated against a wide range of pathogens. Preventing diseases by improving the immunity of fish and shellfish represents a new direction in the development of pollution-free aquaculture, which is important for disease control, health safety, and environmental protection in aquaculture.

Although the advantages and efficacy of herbal medicine in aquaculture have been proven in production practice and have great potential for development in the future, herbal medicine in aquaculture is still in its infancy and has many shortcomings at this stage, mainly manifested in the following aspects: (1) there is no unified standard for medication, because most herbal medicines are not measured by ingredient content, but rely on practical effectiveness accumulated over the years, which is difficult to quantify; (2) most products are crude products. The ingredients are limited by multiple influences such as geography, climate, and time, and the effect is always unstable; (3) the therapeutic effect is not rapid, and the effect on the treatment of acute infectious diseases is not obvious; (4) combined application is not effective. The ratio tests are not enough, and the interactions between medicinal plants of different genera are not clear. Thus, it is difficult to achieve a scientific ratio; (5) commercial production is faced with difficulties, and the technical process of isolation and mass extraction of active pharmaceutical ingredients in medicinal plants is not perfect. Therefore, it is difficult to control low-cost production.

Therefore, future research on the application of medicinal plants in aquaculture should mainly focus on combining medical theory with modern technology, establishing sound techniques for cultivation, extraction, and refinement of medicinal plants, and achieving standardized and commercialized production of new medicinal plant products through improved processing techniques. Moreover, it is necessary to strengthen the research on the specific mechanisms of the role of medicinal plants in aquaculture. It is not enough to focus only on the effect of action, because there are many uncertainties in practical application, and if the specific mechanism of action is not clear, then only the rigid use of medicinal plants is likely to be counterproductive in the end. In conclusion, the effects of medicinal plants in aquatic animals is an exciting topic in aquaculture.

## 4. Conclusions

Medicinal plants have great potential in terms of antivirals; whereas most of the current studies on the antiviral effects of medicinal plants focused on the inhibitory effects of plants on viruses, relatively few established a clear understanding of the antiviral mechanisms of the active pharmaceutical ingredients. In addition, few studies on the synergistic and antagonistic effects of different medicinal plants have been reported. Future research work should focus on the basic research of medicinal plants to clarify their active pharmaceutical ingredients, as well as their pharmacological and toxicological effects; accordingly, they can then be better applied in the prevention and treatment of aquatic animal diseases

## Figures and Tables

**Figure 1 viruses-14-01281-f001:**
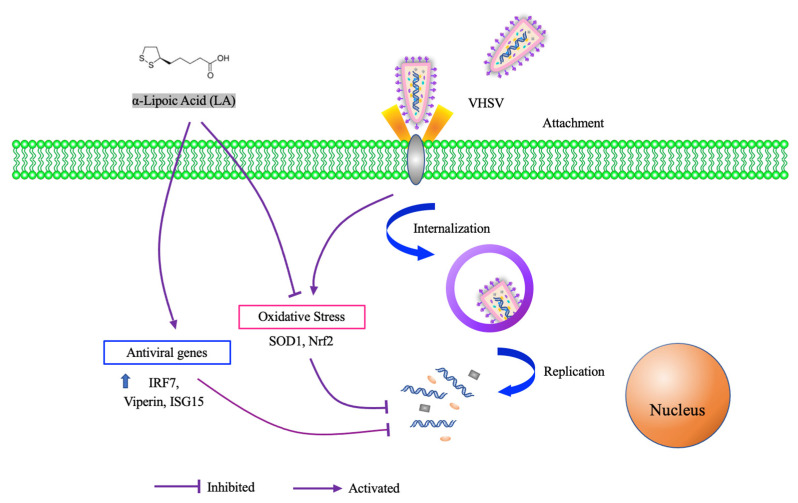
Mechanism of VHSV replication inhibition by α-lipoic acid.

**Table 1 viruses-14-01281-t001:** Medicinal plants and active compounds exhibiting antivirus activity in fish or shellfish and their effects or possible mechanisms of action.

Pathogenic Viruses	Medicinal Plants	Active Compounds	The Possible Mechanisms	References
IPNV	*Heliotropium filifolium*	Filifolinyl senecionate	Inhibiting the synthesis of viral genomic RNA	[28]
	/	Mycophenolic acid (MPA)	Restraining cellular GMP synthesis by inhibiting IMPDH; disturbing RNA polymerase	[30]
IHNV	/	Carotenoid	Improving nonspecific immunity	[42]
	*Lentinus edodes mycelia*	Lentinan (LNT-I)	Regulating the innate immune response and specific immunity	[43]
	*Prunella vulgaris* L. (PVL)	Ursolic acid (UA)	Inhibiting virus replication	[45]
	*Rhus verniciflua* Stokes	Flavonoids	Inducing apoptosis of cells	[47]
VHSV	/	Mycophenolic acid (MPA)	Restraining cellular GMP synthesis by inhibiting IMPDH; disturbing RNA polymerase	[30]
	*Rhus verniciflua* Stokes	Flavonoids	Inducing apoptosis of cells	[47]
	*Celosia cristata* and *Raphanus sativus*	Extract	Inducing gene expression involved in the innate immune response	[57]
	Licorice (*Glycyrrhiza uralensis*, GUF)	Extract; glycyrrhizin (GL); glycyrrhetinic acid (GLA)	Inhibiting the early fusion steps	[59]
	/	Curcumin	Inhibiting virus entry in cells	[61]
	Olive tree leaf *(Olea europaea)* (LExt)	Extract; oleuropein (Ole)	Inactivating virus particles; inhibiting cell-to-cell spread	[63]
	/	α-Lipoic acid (LA)	Inducing antiviral gene expression; reducing VHSV-induced oxidative stress	[65]
SVCV	/	β-Glucose	Regulating the innate immune response	[76]
	*Herbaspirillum camelliae*	Selenium	Activating IFN-related gene expression	[77]
	*Astragalus membranaceus*	*Astragalus* polysaccharide (APS)	Activating IFN-related gene expression	[80]
	*Astragalus membranaceus*	APS	Stimulating the immune response of host; reducing SVCV-induced apoptosis	[150]
	/	Palmitic acid (PA)	Inhibiting autophagy	[81]
	*Psoralea corylifolia*	Bavachin (BVN)	Blocking SVCV-induced apoptosis	[84]
	*Bupleurum yinchowense*	Saikosaponin D (SSD)	Reducing SVCV-induced apoptosis	[85]
NNV	*Gymnema sylvestre*	Gymnemagenol	Inhibiting virus replication	[94]
	/	GLS inhibitor	Involved in regulation of the TCA cycle	[95]
	/	β-Glucan	Stimulating the innate immune memory of macrophages	[96]
GCRV	*Magnolia*	Extract; magnolo; honokiol	Facilitating the expression of innate immune-related genes; restraining GCRV-induced apoptosis	[104,105]
	Green tea	Extract; EGCG	Inhibiting viral particle adhesion to cells	[107]
	Green tea	ECG	Inhibiting viral particle adhesion to cells	[109]
	*Panax ginseng*	Ginsenoside Rg3	Activating IFN-related gene expression	[110]
	/	Quercetin (Qct)	Counteracting the pro-viral effect of heat-shock response	[113]
SGIV	*Illicium verum* Hook. f.	IVAE; IVEE; DDBA; Qct	Inactivating virus particles; inhibiting early viral entry phases; inhibiting virus replication	[122]
	*Illicium verum* Hook. f.	Qct	Promoting the recognition of SGIV and activating the IFN pathway	[123,124]
	*Curcuma kwangsiensis*	CKEE; urdione	Inhibiting virus replication	[126]
	*Lonicera japonica* Thunb.	IAA, IAB, IAC, CA, LT, IS	Inhibiting virus replication	[127]
	*Viola philippica*	Extract	Disturbing virus binding, entry, and replication in host cells	[129]
	*Glycyrrhiza uralensis* (GUF)	Extract	Impacting the binding of virus particles to cell receptors and the replication of viruses in host cells	[130]
WSSV	Green tea	EGCG	Inducing gene expression involved in the innate immune response	[138,139]
	*Typha angustifolia*	Naringenin (NAR)	Restraining early viral gene replication	[140]
	Pericarpium Citri Reticulatae	Hesperidin	Improving nonspecific immunity	[142]
	*Gardenia jasminoides*	Extract	Blocking viral immediate-early stage gene transcript	[143]
	*Gardenia jasminoides*	Genipin (GN)	Attenuating oxidative stress and inflammatory; decreasing signal transducer and activator of transcription gene expression	[151]
	*Eucommia ulmoides*	Geniposidic acid (GPA)	Restraining early viral gene replication; promoting apoptosis	[144]
	*Pongamia pinnata*	Bis(2-methylheptyl)phthalate	Improving nonspecific immunity	[146]
	*Gracilaria tenuistipitata*	Extract	Enhancing the innate immunity	[148]
	*Kappaphycus alvarezii*	Carrageenan	Improving nonspecific immunity	[152]
	*Echinacea purpurea; Uncaria tomentosa*	Extract	Increasing the activity of phenoloxidase	[153]
	*Argemone mexicana*	Extract	Inhibiting viral multiplication; stimulating immune system	[154]
	Mixture of garlic, echinacea, ginger, and basil	Powdered plants	Improving nonspecific immunity	[155]
	*Lonicera japonica*	Luteolin (LUT)	Inhibiting the expression of important viral genes; enhancing antioxidant defenses; mitigating inflammation; inducing apoptosis	[156]
	/	Chicory polysaccharides (CP)	Enhancing antioxidant activity; enhancing anti-WSSV resistance	[157]
	/	Quercetin	Regulating the innate immune response	[158]
	*Psidium guajava*	Extract	Improving nonspecific immunity	[159]
	Lemon; orange	Hesperetin	Regulating the innate immunity	[160]
	*Agathi grandiflora*	Extract	Enhancing antioxidative enzyme gene expression	[161]
	Mixture of *Cyanodon dactylon*, *Aegle marmelos*, *Tinospora cordifolia*, *Picrorhiza kurooa*, and *Eclipta alba*	Extract	Improving nonspecific immunity	[162]
	*Cynodon dactylon*	Extract	Preventing the entry of the virus into the host; preventing the multiplication of the virus in the host cell; enhancing the innate immunity	[163,164,165]
	*Aloe vera*	Powdered whole leaf	Improving nonspecific immunity	[166]
	*A. marmelos*; *C. dactylon*; *L. camara*; *M. charantia*; *P. amarus*	Extract	Improving nonspecific immunity	[167]
	*Uncaria tomentosa*	Extract	Scavenging free radicals; increasing the activity of phenoloxidase	[168]
	*Sonneratia alba*	Extract	Reducing the destruction of blood cells by viruses	[169]
	/	Glycerol monolaurate (GML)	Increasing hemocyte apoptosis, total hemocyte count (THC), PO, and SOD activity; enhancing the expression of immune-related genes	[170]
	*Olea europaea*	Extract	Reducing WSSV-induced oxidative stress	[171]
	*Anoectochilus roxburghii*	*Anoectochilus roxburghii* polysaccharides (ARPs)	Upregulating the expression level of multiple immune genes; promoting the apoptosis of hemocytes	[172]
	*Ophiopogon japonicus*	Extract	Blocking early gene transcription; inducing cellular autophagy; attenuating WSSV-induced oxidative stress	[17]
	Green tea	EGCG	Inhibiting virus replication	[173]
	/	Esculin	Attenuating the infectivity of viral particles; increasing the expression of antimicrobial peptides (AMPs)	[174]
	*Paeonia lactiflora*	Paeoniflorin	Improving nonspecific immunity, especially by increasing the expression of AMPS	[175]
	*Gardenia jasminoides*	Geniposide (GP)	Restraining early and late viral gene expression	[176]
	*Hizikia fusiforme*	Extract	regulates the innate immunity	[177]

## Data Availability

The data that support the findings of this study are available from the corresponding author upon reasonable request.

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
