# Peer review of "Review of Medicinal Plants and Active Pharmaceutical Ingredients against Aquatic Pathogenic Viruses"

_viruses, 2022, doi:10.3390/v14061281_

Round 1

Reviewer 1 Report

The manuscript is more well-written and easier to read and comprehend. Thank you for considering my suggestions. I have just a few more observations:

* Line 76: Latin name in italics

* Line 90: viral diseases prevention (not virus prevention)

* Line 615: "in the final stage of the disease occur cell... and eventual disease shrimp death" 

* Line 703: "speculation" is not a good word. Maybe "theorise"? 

* Line 744: "autoimmunity" - change to "immunity"

Success!

Reviewer 2 Report

The author revised the manuscript. However, there are still some modifications to be made before this article is officially published.

1.      Firstly, the author did not collect the literature carefully. For instance, “4′-(8-imidazole-octyloxy)-arctigenin efficiently inhibits spring viraemia of carp virus infection in vitro and in vivo”, “Synthesis and biological evaluation of novel coumarin derivatives in rhabdoviral clearance”, “Dietary Hizikia fusiforme enhance survival of white spot syndrome virus infected crayfish Procambarus clarkia”, “Highly efficient inhibition of infectious hematopoietic necrosis virus replication mediated by a novel synthesized coumarin derivative in vitro and in vivo, “Evaluation on antiviral activity of a novel arctigenin derivative against multiple rhabdoviruses in aquaculture”, “Antiviral activity of esculin against white spot syndrome virus: A new starting point for prevention and control of white spot disease outbreaks in shrimp seedling culture”. These are the natural compounds and their derivatives which have been reported to have anti-aquatic virus activity. However, it’s not mentioned in the manuscript.

2.      The author has added relevant literature to the text without amending it in the appropriate place in Table 1.

3.       Some of the drug antiviral mechanisms listed in Table 1 cannot be considered mechanisms. For example, “Inhibiting virus replication”. Anti-viral drugs definitely inhibit the replication of the virus. This is a phenomenon, not a mechanism. The author needs to revise form 1 and the relevant statements in the paper.

4.      Line 179 When the abbreviation stands for genes, it should be in italics.

Reviewer 3 Report

The manuscript is well written

This manuscript is a resubmission of an earlier submission. The following is a list of the peer review reports and author responses from that submission.

Round 1

Reviewer 1 Report

The review has been very well written and presented.

Reviewer 2 Report

This study introduces the medicinal plants and their active ingredients for preventing and treating aquatic viruses, and expounds their possible antiviral mechanism, which could be used as a reference for the aquaculture industry. However, the article has some problems that need to be corrected.

  1. line 5 What is the research institution that d stands for?
  2. The logic of the introduction written by the author needs to be revised. My suggestion is that the author should directly introduce the increasing importance of aquaculture, and viral diseases are a major problem plaguing aquaculture. At the same time, the abuse of antibiotics is a side effect of the outbreak of aquatic viral diseases. Therefore, medicinal plants were selected, the advantages of herbal medicine and its application are discussed. The prospect of its application in aquatic virus is introduced at last.
  3. The application of herbal medicine in aquaculture should be quoted more in the introduction to show its application value.
  4. The authors did not update the current literature. For example, “Potential application of antiviral coumarin in aquaculture against IHNV infection by reducing viral adhesion to the epithelial cell surface”, “Synthesis and biological evaluation of novel coumarin derivatives in rhabdoviral clearance”, “Therapeutic potential of phenylpropanoid-based small molecules as antiSVCV agents in aquaculture” “Rhabdoviral clearance effect of a phenylpropanoid medicine against spring viremia of carp virus infection in vitro and in vivo”, “Natural ingredient paeoniflorin could be a lead compound against white spot syndrome virus infection in Litopenaeus vannamei”, “Difference in medication pattern potentially enhances antiviral efficiency of a novel amino-fluorophenyl compound on WSS risk in shrimp seedling culture”, “Small Molecule Inhibitors of White Spot Syndrome Virus: Promise in Shrimp Seedling Culture” and so on.
  5. The antiviral mechanism of these drugs summarized by the authors is not thorough and accurate. I think the author should discuss the antiviral mechanism of drugs in depth.
  6. These viruses summarized by the author should be classified, such as written in the order of DNA/RNA.
  7. In the perspectives part, the author points out many disadvantages of herbal medicine. However, specific solutions are not given. Here the author should discuss more about how to solve these problems.

Reviewer 3 Report

I liked reading the review. I was indeed studying medicinal plants as a potential treatment against WSSV in shrimpBelow are some suggestions that I hope can help improve the quality of your review even more 

  • The review has a large quantity of data, the authors certainly were very dedicated to organizing all the data. However, most of them are a compilation of other studies. What we expect from a review is the debate around all the data, not only the compilation. Please, take the time to better discuss all the information you have organized, presenting a more depth discussion.  

  •  
  • The authors use a considerable number of acronyms throughout the text. I did not check all of them, but there is a chance the acronyms are repeated but with different meanings. It would be useful to present a list of acronyms at the very beginning of the text and be sure they are not used twice or even more. 

  •  

  • Always present the Latin name of any plant / animal / algae species you cite. I started highlighting them, but there were many. Please, revise this. 

  •  
  • Similarly, sometimes the authority's name appears after the Latin name, but most of the time the authority is not shown. Please, standardize. Also, standardize the way the authority's name appears, for example, L., Linn., Linnaeus. 

  •  
  • Be more specific when recommending a given herb: present the recommended dose and time of treatment, if available.  

  •  
  • Revise the grammar. The manuscript has some minor errors in this regard. You may want to use Grammarly, a free software.  

  •  
  • Introduction: in the very first paragraph, it is not clear if you are talking only about human medicine or both human and animal medicine. Please, make it clear.  

  •  
  • Line 72: “human microbiota”, instead of “human flora”. Please, check throughout the text and avoid using “flora” as it refers to flowers, a better term is “microbiota”. 

  •  
  • Line 78: “growth of fish and shellfish”. 

  •  
  • Line 80: present the Latin name of ginseng, at least the genus.  

  •  
  • Line 108: remove the comma. 

  •  
  • Line 128: RT-qPCR: “q” in italics. Check the whole text. 

  •  
  • Lines 176, 188: which species of fish? When citing a species, be specific with the species.  

  •  
  • Sometimes in vivo and in vitro are in italics, other times not. Please, standardize. 

  •  
  • Line 246: which Korea? South Korea? 

  •  
  • Line 258, 260:  “sativus” in italics 

  •  
  • Lines 269 – 278: algae are not plants. You may want to start the sentence with this idea. Despite not being plants (as the core of your review), algae also have medicinal value for the aquaculture industry. 

  •  
  • Sometimes the common name of the plant is cited, which I think is of great importance as the reader may not be familiarized only with the Latin name. You may want to present the common name of the plant examples, as you did on Lines 279, 302. 

  •  
  • Line 289: Latin name of curcumin. The Latin name must be present every time you cite a plant/herb/algae. 

  •  
  • Lines 331-312: Latin name in italics. 

  •  
  • Line 319: present the Latin name. Please, check throughout the text for the absence of Latin names. 

  •  
  • Figure 1 is not cited in the main text. Please, refer to it in the main text and explain it. 

  •  
  • Line 342: “serious pathogenic pathogen” is redundant. 

  •  
  • How SVCV can be highly tolerated (line 351) if it is highly lethal (line 343)? 

  •  
  • Lines 364 – 368: add the reference. 

  •  
  • Line 381: “in vitro and in vivo” are without context. 

  •  
  • Line 382: add the Latin name. 

  •  
  • Line 386: what is a “proper selenium”? Why not only “selenium”? 

  •  
  • Line 391: italics on Latin name 

  •  
  • Line 395: “four” instead of “4”. 

  •  
  • Lines 454 - 464: “Gymnemageno” is capitalized or not? Standardize. 

  •  
  • Lines 476 – 477: “Office International des Épizooties (OIE)”. Also, is it of compulsory communication? If so, state that.  

  •  
  • Line 488: Reductant and without purpose. Mature virus particles are the same as virions, which are the complete infectious virus particle. Like all other viruses to be infectious, they need to be virions. 

  •  
  • Line 489: what is a “double coat” virus? Double capsid?  

  •  
  • Line 492: Latin in italic. 

  •  
  • Lines 584 – 586: this is the purpose of a vaccine: prevention, not treatment. This is what is expected to be done and to occur. 

  •  
  • Lines 647 – 648:Shrimp infected with WSSV cause a large number of deaths”. It seems the shrimp is causing the death. Please, rephrase. 

  •  
  • Line 654: what is “EGCC”? Please, define the acronym. 

  •  
  • Line 657: you did not mention crabs as susceptible to WSSV in the introductory part of the WSSV section. Like crayfish on line 676. 

  •  
  • Line 701: inhibition of what? WSSV replicate?  

  •  
  • Line 701: cite references for further studies. 

  •  
  • Line 712: remove the “...” 

  •  
  • Line 724: “spontaneous immune system”? Is this a new term in immunology? Also, text between lines 723 – 729 gives the impression that the shrimp's innate immune system is not strong/good enough, which is erroneous. Please, rephrase.  

  •  
  • Table 1: unless I missed it, table 1 is not cited in the text. 

  •  

  • Line 735: “sustainable food industry”. 

  •  
  • Lines 744 – 747; 747 – 749; 752 - 753: please, add references. 

  •  
  • Line 754: fish and shellfish. 

  •  
  • Line 754: “autoimmune” is not an appropriate word in this context. Please, change. 

  •  
  • Line 767: what is “genuses”? Genus? 

  •  
  • Line 765 – 768: Particularly, I have a different point of view, as the synergic effect of different plants/herbs can boost their potential medicinal effect. It is important to isolate and study the different compounds but working with the whole plant and even synergies is also a very interesting approach. 

Thank you and success!